# pH-Responsive Super-Porous Hybrid Hydrogels for Gastroretentive Controlled-Release Drug Delivery

**DOI:** 10.3390/pharmaceutics15030816

**Published:** 2023-03-02

**Authors:** Ajkia Zaman Juthi, Fenfen Li, Bo Wang, Md Mofasserul Alam, Md Eman Talukder, Bensheng Qiu

**Affiliations:** 1School of Life Science and Medicine, Department of Biochemistry and Molecular Biology, University of Science and Technology of China, Hefei 230027, China; 2Center for Biomedical Imaging, University of Science and Technology of China, Hefei 230026, China; 3School of Pharmacy, Department of Pharmaceutics, China Pharmaceutical University, Nanjing 211198, China; 4School of Chemistry and Chemical Engineering, Hefei University of Technology, Hefei 230026, China; 5Guangzhou Institute of Advanced Technology, Chinese Academy of Sciences, Guangzhou 511400, China

**Keywords:** super-porous hybrid hydrogels, gastroretentive, controlled-release drug delivery

## Abstract

Super-porous hydrogels are considered a potential drug delivery network for the sedation of gastric mechanisms with retention windows in the abdomen and upper part of the gastrointestinal tract (GIT). In this study, a novel pH-responsive super-porous hybrid hydrogels (SPHHs) was synthesized from pectin, poly 2-hydroxyethyl methacrylate (2HEMA), and N, N methylene-bis-acrylamide (BIS) via the gas-blowing technique, and then loaded with a selected drug (amoxicillin trihydrate, AT) at pH 5 via an aqueous loading method. The drug-loaded SPHHs-AT carrier demonstrated outstanding (in vitro) gastroretentive drug delivery capability. The study attributed excellent swelling and delayed drug release to acidic conditions at pH 1.2. Moreover, in vitro controlled-release drug delivery systems at different pH values, namely, 1.2 (97.99%) and 7.4 (88%), were studied. These exceptional features of SPHHs—improved elasticity, pH responsivity, and high swelling performance—should be investigated for broader drug delivery applications in the future.

## 1. Introduction

Super-porous hydrogels have an inter-correlated arrangement with analogously tiny pores and an intensive swelling rate [1]. Second-generation hydrogels can deliver the required gastric retention application with considerably higher strength. Third-generation hydrogels, that is, super-porous hybrid hydrogels (SPHHs), have been at the forefront of the development of excellent mechanical stretching properties. However, there was a need to develop SPHHs with not only flexibility, but also adequate mechanical stability [2]. Moreover, the SPHHs dosage form extends gastric residence time by targeting site-specific drug release in the stomach’s upper gastrointestinal tract (GIT), conducting gastroretentive drug delivery (GrDD) [3,4]. Hence, the swelling rate of SPHHs is impeded in the blood because of the blood’s weak wetting of the dry hydrogel and its inadequate thickness. Therefore, acidic SPHHs can swell in the stomach through GrDD. The oral drug delivery system is the most convenient treatment method. Recently, there has been more of a drift toward creating novel drug conveyance frameworks that enhance bioavailability and therapeutic selectivity [5]. Controlled-release drug delivery (CrDD) is an adequate human body treatment and the most active scientific research area [6]. On the other hand, interpenetrating networks (IPNs) can be swollen by incorporating two or more monomers or co-polymers using a binder or in situ co-polymerization [7]. Semi-IPNs can have two different compositions (base polymer and linker) because of cross-linking and non-cross-linking through co-polymerization [8,9]. The gas-blowing technique is widely used in the design of gastroretentive SPHHs with a semi-IPN and polymerization arising from gas bubbles for GrDD.

Pectin consists of carboxylic groups that participate in esterification using methanol. It is an anionic biopolymer that is a water-soluble and a pH-sensitive polymeric material in character [10,11,12,13,14,15]. Moreover, poly 2-hydroxyethyl methacrylate (2HEMA) is used to construct an extensive polymer-based arrangement. Therefore, pharmaceutical and biomedical researchers have concentrated heavily on applications (such as inserts, contact focal points, cell immobilizers (drug conveyance carriers), etc.) [16]. In addition, N, N methylene-bis-acrylamide (BIS) is a molecule used as a cross-linking agent that imparts higher strength and toughness to polymer applications and for which there is significant demand in the chemical industry. The literature reveals an absence of data on the effect of a mixture of pectin and 2HEMA-based SPHHs on GrDD [17]. In this study, 2HEMA is used as a binder to enhance mechanical stability [18]. Furthermore, the eradication of the *Helicobacter pylori* (*H. pylori*) bacterium (the gastric mucous layer of the stomach) is specified for gastric ulcers. With the development of gastro-duodenal diseases and the prolonged local application of the drug amoxicillin trihydrate (AT), *H. pylori* infections effectively diffuse bacteria. Amoxicillin is a familiar sub-class of penicillin in clinical remedies. The chemical structure of amoxicillin is different from penicillin in the side chain, which retains further alfa-amino and p-hydroxy groups [19,20].

This research article concentrates on pH-responsive gastroretentive SPHHs for CrDD. SPHHs were prepared from pectin and 2HEMA through cross-linking. The mechanical properties of the hydrogels were developed with the help of semi-IPNs using N, N methylene-bis-acrylamide (BIS). The SPHHs were considered for their swelling properties, mechanical stability, gelation kinetics, density, and drug content, and in vitro drug release studies were performed to define CrDD in acidic conditions.

## 2. Materials and Methods

### 2.1. Materials

Amoxicillin trihydrate (AT), N, N methylene-bis-acrylamide (BIS) (Aladdin industrial corporation), pectin (Sigma-Aldrich Co., Ltd., Burlington, VT, USA), 2-hydroxyethylmethacrylate (2HEMA), pluronic F27/poloxamer 407, ammonium persulfate (APS), N, N, N, N-tetramethylene diamine (TEMED), glacial acetic acid, sodium acetate, and sodium bicarbonate were purchased from commercial sources (China). Acetone and ethanol (AR anhydrous) were acquired from Shanghai Titan Scientific Co., Ltd. (Shanghai, China). Deionized water was available as and when required and utilized throughout.

### 2.2. Synthesis of SPHHs

The stock solution of super-porous hybrid hydrogel 3% w/v was prepared by dissolving pectin in 0.1 M glacial acetic acid and 10% v/v AQ 2HEMA solution. The pectin and 2HEMA solutions were mixed with different compositions [21]. Each pectin and 2HEMA mixture was placed in a test tube. The pH was adjusted to around 4.5–5 with a 5 M sodium hydroxide solution [22]. BIS AQ solution (4% w/v) was added to each pectin and 2HEMA mixture. The following substances were added to the test tube at 60 °C: 10% w/v pluronic F127/poloxamer 407, 2% v/v TEMED, and 1% w/v APS AQ solution [23]. For efficient mixing, the polymerization process was operated for 10 min, and as substances were added to the test tube the reaction caused intensive shaking (Appendix A) [24]. In the end, 70 mg of NaHCO₃ was added to the solution and gently mixed, which produced gel and foaming reactions [25] (Appendix A).

### 2.3. Drying of SPHHs

Swollen super-porous hybrid hydrogels were dried by applying around 8 mL of ethanol per gel. After this underlying lack of dehydration step, SPHHs were dried out further by putting them in 60 mL ethanol for a few minutes to remove water. After the dehydration, the abundance of ethanol in dried-out SPHHs was expelled by depleting the filter paper (Appendix A). At that point, the SPHHs were dried in an oven at 60 °C for 24 h [22]. The dried hydrogels were kept in a desiccator until further characterization.

### 2.4. Drug (Amoxicillin Trihydrate) Loading of SPHHs-AT

The soaking technique was performed for the drug loading [26]. The swelling of the SPHHs was accomplished by adding AT in DMSO (dimethyl sulfoxide) solution and a buffer solution of glacial acetic acid to adjust the pH to 5 [26] (Appendix A). Herein, we exposed the SPHHs to the drug solution bottle and waited until the drug solution was absorbed by the SPHHs and dried at 45 °C overnight in an oven dryer [27]. Meanwhile, the drug solution of the required concentration was formulated.

### 2.5. pH-Responsive Swelling of SPHHs-AT

The mechanical properties of the excellent swelling performance of SPHHs-AT were sensitive. As we know, stimuli factors such as ionic strength, pH, salts, pressure, and organic solvent also influence the swelling. Therefore, the swelling was measured gravimetrically and volumetrically, and a surface was utilized to determine the SPHHs-AT swelling features under a load [28]. Additionally, swelling parameters were assessed at low or average or a raised body temperature of 37 °C. The pH-subordinate of the SPHHs-AT was assessed at 37 °C with a variation of the swelling condition between HCl and phosphate buffer solution at pH 1.2 and 7.4, respectively [29]. The SPHHs-AT was first swollen in HCl solution for 35 min, weighed, and immersed in PBS and vice versa. A graph was created of the swelling ratio vs. time (min). The following equation (Equation (1)) calculates the swelling ratio (Q_s_):(1)$$Q_{s}=\left(W_{s}-W_{d}\right)/\left(W_{d}\right)\times 100$$
where W_s_ is the weight of wet SPHHs and W_d_ is the weight of dried SPHHs.

### 2.6. Density of SPHHs

The dissolvable technique was utilized for thickness measurement. The dried SPHHs were treated with various solvents for density estimations [30]. A small number of SPHHs were taken, and their density was calculated via Equation (2):(2)$$\mathrm{Density}\;\left(\rho \right)=\frac{M_{\mathrm{SPHHs}}}{V_{\mathrm{SPHHs}}}$$
where M_SPHHs_ and V_SPHHs_ represent the mass and volume of the SPHHs [30].

### 2.7. Gelation Kinetics of SPHHs

The gelation time was depicted as a term for gel formation and estimated by a simple tilting technique after pH adjustment to 5. The time reactant blend specified how long it took to become viscous, and the solution no longer dropped in the tilted tube position. During the polymerization process, the solution’s viscosity increased until the gel structure formed [3]. Herein, the gel structure was determined using the length of time, and viscous liquefaction was used for a unique shape [31].

### 2.8. Mechanical Stability of SPHHs

The mechanical intensity of the SPHHs was viewed using a modified procedure employing digital hardness testers (NANOVEA Mechanical Tester). A swollen sample was placed and force applied until SPHHs breakage [32]. The sample cross-sectional range determined the definitive intensity.

### 2.9. SEM

The field emission scanning electron microscope (SEM) (Schottky–ZEISS Gemini SEM 450) was utilized to capture images using a digital capture card and Digital Scan Generator [33]. The SEM was used to view the dried SPHHs surface morphology as the sample was cut into transverse sections and mounted on double-sided tape on aluminum stubs [34].

### 2.10. FTIR^ATR^, X- RD and DSC

FTIR was analyzed to ascertain the compatibility between selected drugs and SPHHs-AT. The SPHHs-AT FTIR spectra were analyzed using the KBr pellet (Tensor 27 infrared spectrophotometer) and recorded over the range of 400–4000 cm^−1^ [35]. The X-RD spectra were studied to monitor the significant crystallinity aspects of the drug loaded into the hydrogels polymeric network. The freeze-dried, loaded SPHHs-AT was analyzed by the X-RD patterns (D8 X-ray powder diffraction) using the Ni-filtered, CuK α radiation with 40 mA current and a voltage of 45 kV [36]. Differential scanning calorimetry (DSC) was performed on pure drug and SPHHs-AT samples (DSC600 Linkam Scientific Instruments Ltd., Salfords, UK). Under a nitrogen atmosphere, the energy of SPHHs-AT was measured as J/K; also, dynamic scans were obtained toward a heating rate of 10 °C/min. Calorimetric measurements were fabricated with the empty cell (high-purity alpha-alumina discs).

### 2.11. Drug Content and In Vitro Drug Release

Super-porous hybrid hydrogels contained a 50 mg AT weight in a 100 mL flask and were diluted with pH 1.2 HCl buffer [22]. The soluble mixture was filtered, and drug content was determined using a UV–Vis spectrophotometer (UV-2000 spectrophotometer, Kohl Equipment Co., Ltd., Rodermark, Germany) at 272 nm (Appendix A). The loaded drug contents (mg/g) of hydrogel samples were determined using Equation (3).
(3)$$\mathrm{Loaded}\;\mathrm{drug}\;\mathrm{content}\;\left(\frac{\mathrm{mg}}{g}\right)=\frac{\mathrm{Weigh}\;\mathrm{of}\;\mathrm{drug}\;\mathrm{in}\;\mathrm{hydrogel}\;\left(\mathrm{mg}\right)}{\mathrm{Weigh}\;\mathrm{of}\;\mathrm{dried}\;\mathrm{crushed}\;\mathrm{hydrogel}\;\left(g\right)}$$

In vitro drug release from a series of SPHHs-AT samples was determined (using USP dissolution apparatus-type II) at 37 ± 0.5 °C at 100 rpm paddle speed in 900 mL of pH 7.4 and pH 1.2 for 14 h [26]. Each sample was withdrawn in a 10 mL volume amount at a pre-determined interval from the dissolution medium after being immediately replaced with fresh medium. All sample filtration was passed through 0.45 µm filter paper and diluted to a suitable pH 1.2 concentration [37].

## 3. Results and Discussion

### 3.1. SPHHs and Swelling Studies

SPHHs were successfully synthesized via simple cross-linking (Appendix A). In short, according to standard formulations, a water-soluble and ionogelling polymer (synthetic or natural) was added during preparation. After preparation, the SPHHs were treated in an ion solution for strength and elasticity. In the semi-interpenetrating networks of the SPHHs, N, N methylene-bis-acrylamide was applied via chemical cross-linking [38] to generate a bubble in each sample that produced a porous structure. APS and TEMED are commonly used polymerization independent initiators and catalysts [16]. The selective SPHHs-AT samples were used in GrDD with excellent drug loading at pH 5 and released 98% in acidic conditions (pH 1.2) (Figure 1). Figure 1 exhibits the probable synthesis ball-and-stick model of SPHHs and drug (AT) loading into SPHHs at pH 5 and release at pH 1.2 in acidic conditions. Therefore, the innovative synthesized SPHHs performed successfully in CrDD.

The effective property of SPHHs-AT is its rapid swelling ability [39,40]. The swelling ratio of SPHHs-AT preparations in an HCl buffer solution of pH 1.2 is represented in Figure 2a. We studied the effect of BIS, 2HEMA, and pectin on the swelling capability from the results. The ratio of SPHHs-AT was reduced with rising cross-linking density, as much tighter networks and fewer pores were formed at higher concentrations of the cross-linking agent (Figure 2b). The SPHHs-AT was shown to be relatively uniform because of high viscosity during gelation, leading to smaller pores. The swelling ratio of SPHHs-AT decreased owing to the increased 2HEMA monomer concentration during formation [41]. Furthermore, the more complex networks were fabricated as pores closed with the 2HEMA polymer, reducing the flexibility of polymeric chains and retarding their swelling. The swelling ratio of the samples was in the following order of preparations as SPHHs-AT-3 > 2 >1 > 4 > 5 > 6 > 7, because of pH sensitivity and the time duration of samples (Appendix A).

The effects of the swelling reversibility studies of SPHHs-AT between pH 1.2 and 7.4 solutions are illustrated in Figure 3. The structure of the SPHHs-AT with large numbers of pores connected to form capillary channels was advantageous for easy diffusion of the swelling medium into the polymeric matrix, thus contributing to its speedy response to pH change [39]. They rapidly absorbed and released the swelling medium upon the pH alteration from acidic to primary conditions and vice versa (Appendix A). Even after many swelling–de-swelling cycles, the SPHHs-AT remained intact, ensuring its fundamental integrity despite changes in the pH of the external environment [42].

### 3.2. Density and Gelation Kinetics of SPHHs

The apparent density values of the various SPHHs preparations were determined (Figure 2(ci)) [30]. The prepared SPHHs were found between 0.50–0.844 g/cm^3^; since the SPHHs were very porous this was defined as the apparent density (Appendix A). The density increased with the increase in BIS concentration. The presence of cellulosic fibers was noted in the polymer structure. The lower density values indicated an interpenetrating porous structure in the SPHHs [30]. The gelation kinetics provided good evidence for determining the introduction period of the blowing agent (NaHCO₃). NaHCO₃ must be introduced to produce tremendous and uniform pores when the reactants have appropriate viscosity. Bubbles cannot sustain their outline for an extended period if a gas-blowing agent is added too early or if the gelation period is relatively longer. The foaming reaction occurred only below the acidic condition (pH 5.0–5.5), and thus the pH 5 was adjusted [3]. The optimum pH for the gelation was around 7.0–8.0, where the polymerization proceeded speedily and the gelling usually started within 0.5–1.0 min. Hence, NaHCO₃ was introduced 30 s after the adjustment of pH 5. After the addition of BIS, the sol–gel transition time for several formulations was between 24 and 32 s (Figure 2(cii) and Appendix A). This indicated that the blowing agent must be introduced immediately after adding BIS. If the porogen is introduced later, it might lead to a non-porous hydrogel formation [43].

### 3.3. Mechanical Performance of SPHHs

In this study, SPHHs were synthesized as semi-IPNs, leading to an increase in mechanical strength. The stability was improved as compared to earlier reported SPHHs [44]. The presence of 2HEMA in SPHHs resulted in enhanced mechanical properties. Hence, the benefit of modulus polymer networks was realized when associated with conventional SPHHs [44]. The maximum abdominal contraction force of humans is nearly 4900–6800 N/m^2^. The SPHHs were first swollen in the buffer to their equilibrium, and their ultimate compressive strength was determined, see Figure 2(ciii) and Appendix A. Therefore, the amounts of pectin, 2HEMA, and BIS were predominant in defining the SPHHs’ mechanical stability and performance of GrDD [45].

### 3.4. SPHHs’ Morphology

SEM was used to examine the detailed structures inside the SPHHs in the dry states (freeze-dried and vacuum-dried samples) [46,47]. This was conducted to characterize the surface morphology, texture, and porosity of the SPHHs, as shown in Figure 4a–b’. It was detected that drying the SPHHs without the acetone conduct stage resulted in collapsing pores on their surface, whereas drying after treatment with the acetone showed numerous and evenly distributed pores that were linked to each other [30,33,48]. The images showed pores in the structure owing to their high water absorption capacity [49]. Destruction of the porous structure was observed in many places because of its conversion through the grinding method. However, a few pores were not disturbed and visible.

### 3.5. Comprehensive Study of SPHHs and SPHHs-AT

FTIR-ATR was conducted to assess the interaction between the drug and SPHHs-AT-3 as a selective preparation [50]. FTIR spectroscopy was used to ensure the compatibility of the drug and sample and to explore the chemical structure of the synthesized SPHHs. In the AT sample, the broad FTIR spectra of pure AT and its entire characteristic peaks are shown (Appendix A). Between 3528.6 and 2029.8 cm^−1^ and between 1775.8 and 1120.0 cm^−1^ wave numbers were noted. The alkenyl (-C=C-) (3528–2969.6 cm^−1^), amide (-NH) (1775.8–1120.0 cm^−1^), ketone (-C=O) (1020.5–848.0 cm^−1^), and phenolic (-OH) (523.3–538.0 cm^−1^) stretches were mostly responsible for those areas [43,51]. There may be physical connections related to establishing weak to medium-intensity attachment since no key shifting of peaks was noted. Polymers may adjust the rate and pathway of the diffusion of drug molecules by fluctuating entanglement in polymeric networks [49].

Thus, the physical connections might cooperate in sustaining the release of drug molecules from the experimental formulations. An FTIR-ATR spectrum of SPHHs-AT-3 illustrated the characteristic peaks, as shown in Appendix A. DSC studies were carried out for AT, SPHHs, and SPHHs-AT-3 thermograms shown in Appendix A. The pure drug’s thermogram showed a sharp endothermic peak at 95.7 °C, which resembled its melting point [52]. The DSC assessment revealed no interaction between the drug and excipients. DSC assessed the hydrogel’s thermal behavior associated with the hydrogen-bond-induced crystalline structure. As an impact, it was apparent that the melting point of AT did not change when it was formulated as SPHHs-AT-3. DSC was also conducted to ascertain the changes in the SPHHs-AT-3 for increasing cross-linking density. The SPHHs preparations were subjected to DSC studies, which showed the thermal transition performances of the SPHHs-AT-3 preparations as a function of varying amounts of the cross-linking agent [53]. The thermal performances of these SPHHs-AT-3 preparations were examined with DSC because the increased mechanical strength was presumably due to intensified cross-linking density. In Appendix A, it is clear that there was a shift to a higher glass transition temperature with increased BIS. There were no liberally mobile polymeric chains, as most of them would have been cross-linked with an increment in cross-linker volume. Hence, a higher volume of heat energy was required to break the cross-linked chains compared to a loose network.

X-RD analysis illustrates crystalline organic solids comprised of molecules loaded or stacked in a precise sequence. To study the products’ crystallographic nature, the X-RD was performed on two samples (AT and SPHHs-AT-3) (Appendix A). According to the spectra, amoxicillin trihydrate showed a broad peak at 15° because of the amorphous nature of the hydrogel; sharp peaks were observed at 2θ = 12.174, 15.134, 16.254, 17.218, 18.043° [54]. The figure shows the overlay diagram of the X-RD spectra of the drug and SPHHs-AT-3. The gel is much more obvious than the drug, which is why the peak is diffused [55]. However, the drug’s peaks and the reduced diffraction strength of AT suggested a reduction in the crystals’ quality [56].

### 3.6. Drug Loading and Release Performance

The GrDD experiment determined that the drug loading was uniform, and there was an exact distribution of the drug in the SPHHs-AT-(1 to 7) (Figure 2(civ) and Figure 5a,b). The procedure of soaking or equilibration was used for drug loading (a buffer was necessary for the complete swelling of the SPHHs). The SPHHs-AT loading and release are selectively shown in Figure 5. Concisely, the SPHHs performed well in the GrDD because of the interpenetrating porous structure in the SPHHs containing pectin, 2HEMA, and poloxamer 407. These suggested creating a semi-IPN network with coordination environments with drugs in the acidic condition. The probable drug-loading mechanism of SPHHs is shown in Figure 5b. Moreover, the probable drug release mechanism indicating an excellent interaction with the polymer chain with the acidic solution showed 98%, which was the maximum drug released (Figure 5c). In contrast, the novel pH-responsive pectin-based SPHHs were successfully involved in CrDD. The pectin-based SPHHs structure had oxygen-rich carbon bonding with drugs regarding the loading and release function performed in acidic conditions. In acidic conditions, for example, HCl groups coordinated with drug molecules to effect higher release from the SPHHs.

The drug content assessment of dissimilar SPHHs preparations determined that the drug content was in the range of 95.36–98.05% of the total amount of AT in the SPHHs-AT (Appendix A) [49,57]. The drug release originated from the amount of pectin and 2HEMA. It was observed to be inversely related to the amount of BIS. The in vitro drug release studies were carried out as shown ((a) pH 7.4, (b) pH 1.2 in Figure 6a–b’) [58]. The AT release from the SPHHs-AT reflected that the increase in the concentration of BIS and decrease in the volume of pectin extended the release of the drug, with initially fast drug release SPHHs-AT-3 (88%) and delayed drug release SPHHs-AT-7 (50%) for 14 h. The cumulative SPHHs-AT drug released formulations in order were 3 > 2 > 1 > 4 > 5 > 6 > 7.

## 4. Conclusions

SPHHs based on pectin and poly 2-hydroxyethyl methacrylate (2HEMA) were synthesized using the gas-blowing technique. In situ cross-connecting pectin with BIS as a cross-linker might be responsible for the high swelling strength. The compression strength results indicated that semi-IPN SPHHs were mechanically flexible enough to withstand gastric retrenchment and conserve their appearance. The SPHHs were characterized using these methods: swelling capacity, gelation energy, density estimation, mechanical strength, FTIR-ATR, X-RD, and DSC. Then SPHHs was loaded with the drug through an aqueous loading method. The indivisibility of pores was maintained in acetone-treated hydrogels, as examined by SEM This research found that SPHHs-AT could be utilized as an excellent GrDD carrier for AT, given its significant swelling (about 98% at pH 1.2) and delayed drug release attributes in acidic conditions. The drug release was connected to the swelling properties of these polymers. Hence, both polymers could be novel carriers for controlled-release drug delivery. SEM studies indicated the formation of corresponding pores in SPHHs, which were not selected significantly after conversion to particles. The in vitro drug release studies of SPHHs-AT indicated that two different pH levels followed several time intervals (88% and 97.99%). The excellent features of the SPHHs improved the elasticity and pH-responsive fast and strong swelling performance. Pectin-based SPHHs will be researched in diverse pharmaceutical, biomedical, and industrial applications.

## Figures and Tables

**Figure 1 pharmaceutics-15-00816-f001:**
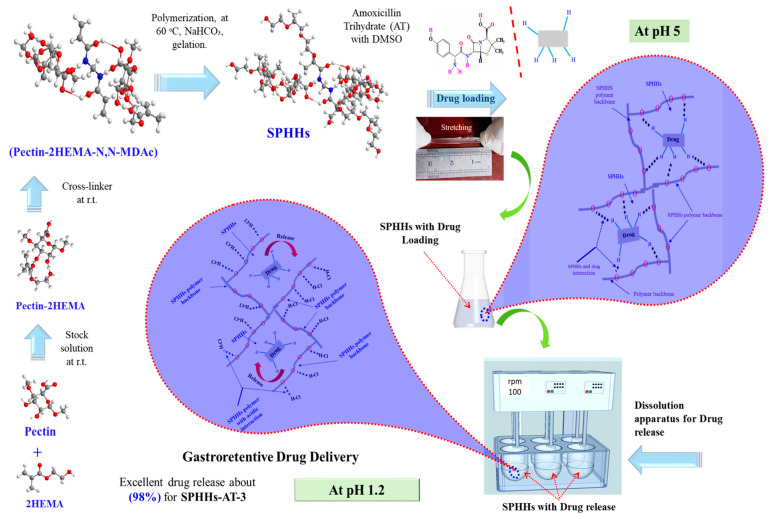
The probable synthesis ball-and-stick model of SPHHs and drug (AT, amoxicillin trihydrate) loading into SPHHs at pH 5 concerning SPHHs-AT and release at pH 1.2 in acidic conditions.

**Figure 2 pharmaceutics-15-00816-f002:**
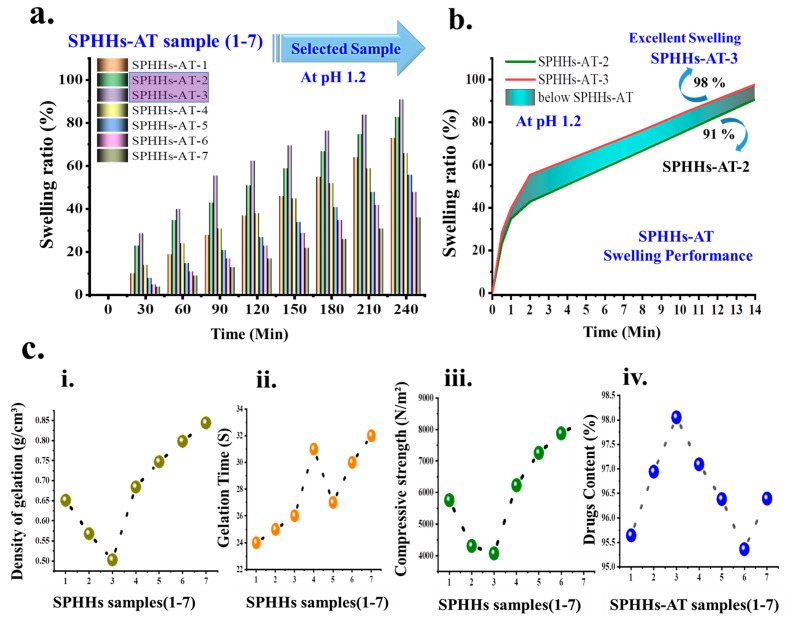
(**a**,**b**) The pH-dependent swelling appearance of SPHHs-AT (pH 1.2) phosphate buffer solutions (PBS) indicates their performance. (**c**) The comparative study of a series of SPHHs samples as (**i**) assigned as apparent density; (**ii**) the performance of gelation kinetics; (**iii**) mechanical strength leading as semi-IPNs; (**iv**) uniform drug content.

**Figure 3 pharmaceutics-15-00816-f003:**
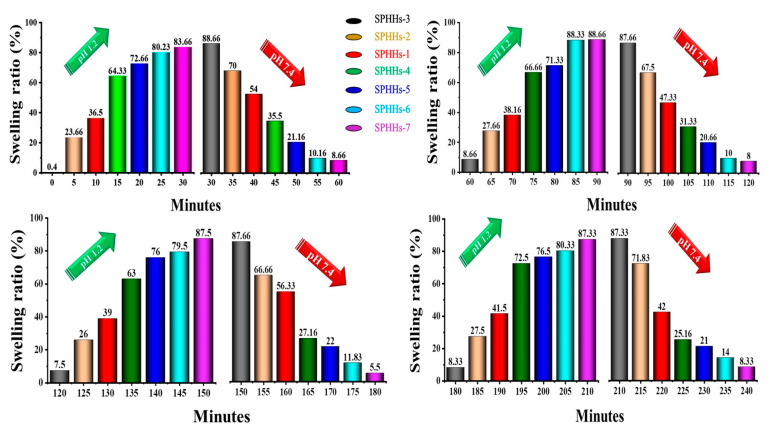
The pH-dependent swelling ratio (%) from the interchange of the medium between (pH 1.2) HCl and (pH 7.4) PBS (*n* = 3, mean ± standard deviation). The swelling ratio versus time measures the different levels of pH absorbed into the SPHHs structure. While this indefinitely reflects the weight of the SPHHs in their dry and fully swollen states, the sample behavior within this period reflects the swelling mechanism.

**Figure 4 pharmaceutics-15-00816-f004:**
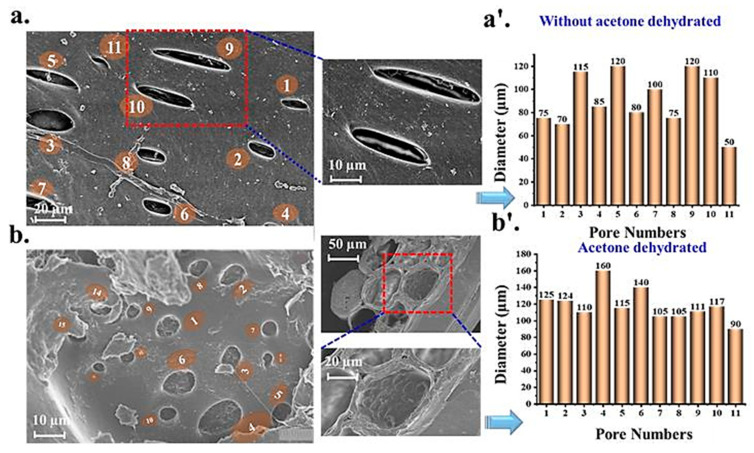
The level of interaction between biofilm and solid-state electrodes, in terms of spatial dispersion, thickness, and morphology, as studied by SEM analysis. SEM photograph of SPHHs. (**a**) Non-acetone dehydrated freeze-dried, single pore of sample; (**a’**) average pore diameter non-acetone dehydrated; (**b**) acetone dehydrated oven-dried, single pore of the sample; (**b’**) average pore diameter, acetone dehydrated.

**Figure 5 pharmaceutics-15-00816-f005:**
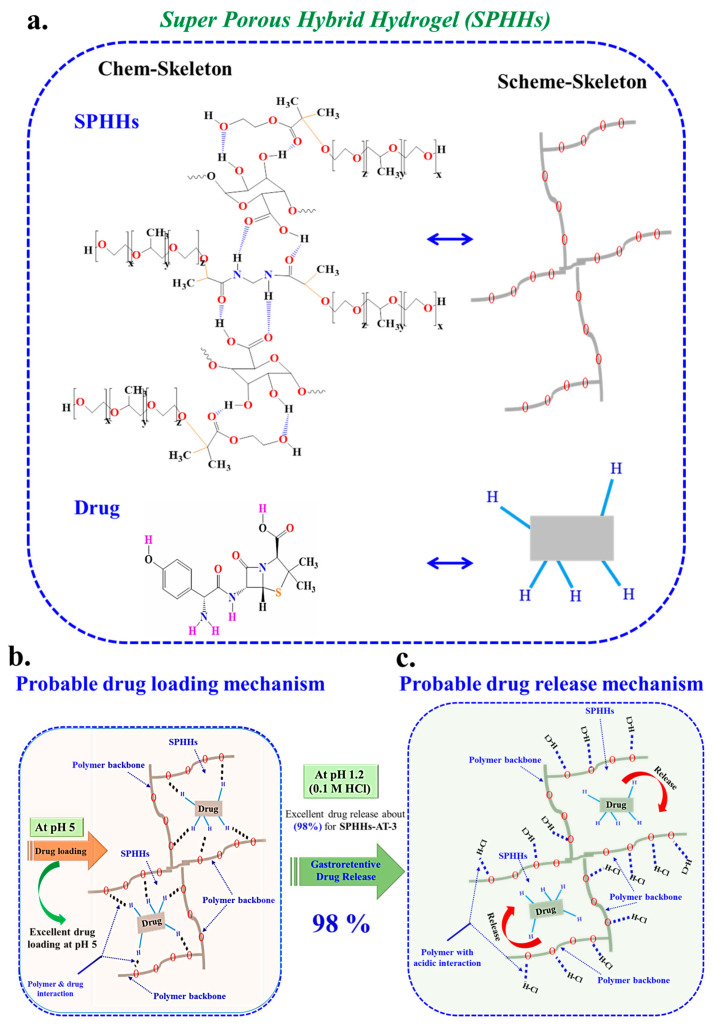
(**a**) The SPHHs structure depends on the chem-skeleton and scheme-skeleton. (**b**) The probable drug-loading mechanism at pH 5 for gastroretentive SPHHs and drug in the chem-skeleton [33]; (**c**) the presumed drug-release mechanism at pH 1.2 for SPHHs-AT [32,55].

**Figure 6 pharmaceutics-15-00816-f006:**
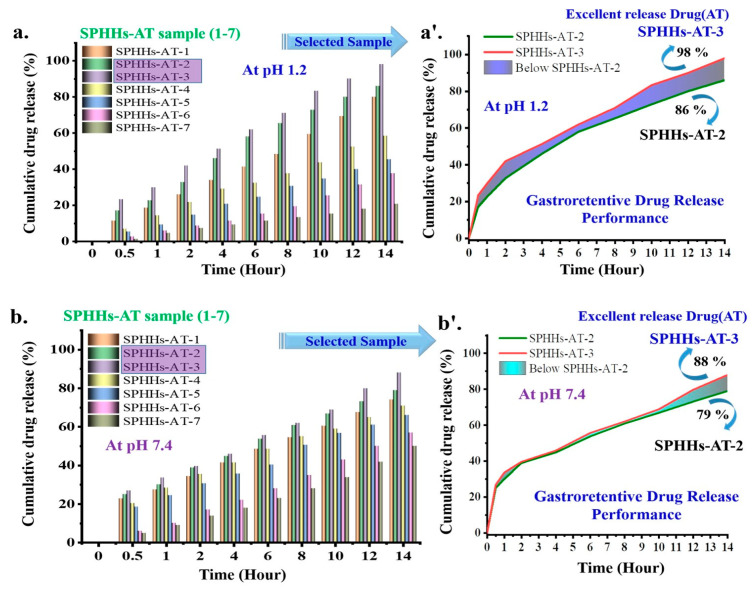
The in vitro drug release profile of AT-loaded SPHHs samples at two different pH values; (**a**,**a’**) pH 1.2 and (**b**,**b’**) pH 7.4, sequentially (*n* = 3, mean ± standard deviation).

## Data Availability

Not applicable.

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
