# Peer review of "pH-Responsive Super-Porous Hybrid Hydrogels for Gastroretentive Controlled-Release Drug Delivery"

_pharmaceutics, 2023, doi:10.3390/pharmaceutics15030816_

Round 1

Reviewer 1 Report

This manuscript deals with “pH-responsive Super Porous Hybrid Hydrogels for Gastrore-tentive Controlled-release Drug Delivery”. The authors prepared SPH materials and then used for drug delivery applications. This work can be accepted for publication after some major revisions as the followings:

1-      Supporting Information, Figure S1: Are all the mentioned bonding are covalent?

2-      Supporting Information, Figure S2: The swelling content by the consumption of NaHNO3 is low. It is required to be an equilibrium between the polymerization and release of CO2 from the reaction of NaHCO3. How did the authors establish this equilibrium? And what is the amount of porosity and swelling of the material? BET or SEM from the cross-section are required.

3-      Figure 6: It seems that a burst release has been carried out at the first 3 h. Can we name this controlled release?

4-      Cell toxicity and viability studies are required.

5-      In vivo investigations are also needed.

6-      I notice some review papers in this subject which totally confirm my comments 4 and 5.

https://www.sciencedirect.com/science/article/abs/pii/S1742706119303356

https://pubs.acs.org/doi/10.1021/acsomega.0c05276

Reviewer 2 Report

Report attached. 

Round 2

Reviewer 1 Report

The authors have revised their manuscript. So, I recommend publication in the current form.

Reviewer 2 Report

The revision is satisfactory and acceptable.